# ADVERSARIAL ROBUSTNESS AS A PRIOR FOR LEARNED REPRESENTATIONS

## ABSTRACT

An important goal in deep learning is to learn versatile, high-level *feature representations* of input data. However, standard networks' representations seem to possess shortcomings that, as we illustrate, prevent them from fully realizing this goal. In this work, we show that *robust optimization* can be re-cast as a tool for enforcing a prior on the features learned by deep neural networks. It turns out that representations learned by robust models address the aforementioned shortcomings and make significant progress towards learning a high-level encoding of inputs. In particular, these representations are approximately invertible, while allowing for direct visualization and manipulation of salient input features. More broadly, our results indicate adversarial robustness as a promising avenue for improving learned representations. [1]

## 1 INTRODUCTION

Beyond achieving remarkably high accuracy on a variety of tasks (Krizhevsky et al., 2012; He et al., 2015; Collobert & Weston, 2008), a major appeal of deep learning is the ability to learn effective representations of data. Specifically, deep neural networks can be thought of as linear classifiers acting on learned feature representations (also known as *feature embeddings*). A major goal in representation learning is for these embeddings to encode high-level, interpretable features of any given input (Goodfellow et al., 2016; Bengio et al., 2013; Bengio, 2019). Indeed, learned representations turn out to be quite versatile—in computer vision, for example, they are the driving force behind transfer learning (Girshick et al., 2014; Donahue et al., 2014), and image similarity metrics such as VGG distance (Dosovitskiy & Brox, 2016a; Zhang et al., 2018).

Still, deep networks' feature embeddings exhibit some shortcomings that are at odds with our idealized model of a linear classifier on top of interpretable high-level features. For example, the existence of adversarial examples (Biggio et al., 2013; Szegedy et al., 2014)—and the fact that they may correspond to flipping predictive features (Ilyas et al., 2019)—suggests that deep neural networks make predictions based on features that are vastly different from what humans use, or even recognize. (This message has been also corroborated by several recent works (Brendel & Bethge, 2019; Geirhos et al., 2019; Jetley et al., 2018; Zhang & Zhu, 2019).) An more direct example of such a shortcoming is pinpointed by Jacobsen et al. (2019), who show that one can find image pairs that appear completely different to a human but are nearly identical in terms of their feature embeddings.

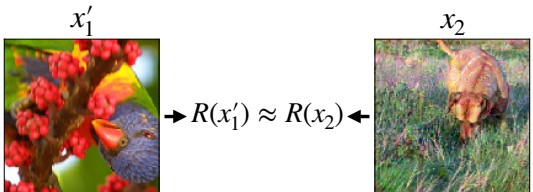

Figure 1: A limitation of standard feature embeddings: it is straightforward to construct pairs of images $(x_1, x_2)$ that appear completely different yet have near-identical representations.

---

[1] https://github.com/cantankerousdolphin/robust-learned-representations.

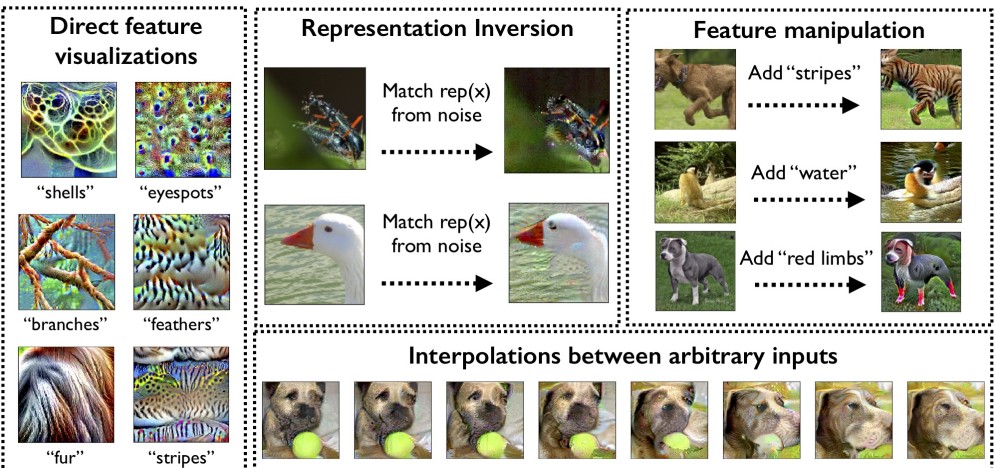

Figure 2: Sample images highlighting the properties and applications of "robust representations" studied in this work. All of these manipulations use only gradient descent on simple, unregularized, direct functions of the representations of adversarially robust neural networks (Goodfellow et al., 2015; Madry et al., 2018).

**Our contributions.** Motivated by the limitations of standard representations, we propose using *robust optimization* (in particular, adversarial training (Goodfellow et al., 2015; Madry et al., 2018)) to enforce a prior on features that models learn (and thus on their learned feature representations). It turns out that the resulting "robust representations" (the embeddings learned by adversarially trained neural networks) are significantly better-behaved than their standard counterparts. We demonstrate this fact by looking at two well-studied tasks that are typically used to study representations:

- **Representation inversion (Section 5)**: In contrast to standard representations—for which representation inversion is a significant challenge, e.g., (Mahendran & Vedaldi, 2015)—robust representations are naturally *approximately invertible*. In particular, adversarially robust networks provide an embedding of the input such that images with similar representations are semantically similar, and the salient features of an image are easily recoverable from its robust feature representation. This property also naturally enables feature interpolation between arbitrary inputs.

- **Feature visualization (Section 6)**: Direct maximization of the coordinates of robust representations suffices to visualize easily recognizable features of the model. This is again a significant departure from standard representations where (a) without explicit regularization at visualization-time, feature visualization often produces unintelligible results; and (b) even with regularization, visualized features in the representation layer are scarcely human-recognizeable (Olah et al., 2017). As a result of this property, robust representations enable the addition of specific features to images through direct first-order optimization.

The tasks above and their respective applications are illustrated in Figure 2. Broadly, our results suggest robust optimization as a promising avenue for learning better-behaved image representations.

## 2 RELATED WORK

Our work studies the feature representations of *adversarially robust networks*. As discussed in Section 3, these are networks trained with the robust optimization framework (Wald, 1945; Goodfellow et al., 2015; Madry et al., 2018) and were originally proposed in the context of defending against adversarial perturbations (Biggio et al., 2013; Szegedy et al., 2014). Adversarial robustness has been studied extensively in the context of machine learning security (see e.g., Carlini & Wagner (2017); Athalye et al. (2018b;a); Papernot et al. (2017)), and as an independent phenomenon (see e.g., Gilmer et al. (2018); Schmidt et al. (2018); Jacobsen et al. (2019); Ilyas et al. (2019); Tsipras et al. (2019); Su et al. (2018).

Several recent works have studied the qualitative properties of robust models. Zhang & Zhu (2019) find that adversarially robust models behave more predictably in the face of various out-of-distribution data, appearing to use more global features. Tsipras et al. (2019) observe that large adversarial perturbation constructed for robust networks actually resemble instances of the target class. Santurkar et al. (2019) leverage this fact and use robust classifiers for a wide array of image synthesis tasks. Our work is complementary to this line of research, and focuses on understanding properties of robust representations through the lens of "benchmark" representation learning tasks (namely, inversion and component visualization).

There is also a large body of work dedicated to studying each of the representation learning tasks we study below (inversion and feature visualization)—we have embedded discussions of the related work for each task within the relevant sections.

## 3 BACKGROUND AND MOTIVATION

### 3.1 ADVERSARIAL EXAMPLES AND ROBUST TRAINING

In standard settings, supervised machine learning models are trained by minimizing the expected loss with respect to a set of parameters $\theta$, i.e., by solving an optimization problem of the form:

$$\theta^* = \min_\theta \mathbb{E}_{(x,y)\sim\mathcal{D}} \left[ \mathcal{L}_\theta(x,y) \right]. \tag{1}$$

We refer to (1) as the *standard* training objective—finding the optimum of this objective should guarantee high performance on unseen data from the distribution.

It turns out that deep neural networks trained with the standard objective are vulnerable to *adversarial examples* (Biggio et al., 2013; Szegedy et al., 2014)—by changing a natural input imperceptibly, one can easily manipulate the predictions of a deep network to be arbitrarily incorrect.

A natural approach (and one of the most successful) for defending against these adversarial examples is to use the *robust optimization framework*: a classical framework for optimization in the presence of uncertainty (Wald, 1945; Danskin, 1967). In particular, instead of just finding parameters which minimize the expected loss (as in the standard objective), a robust optimization objective also requires that the model induced by the parameters $\theta$ be robust to worst-case perturbation of the input:

$$\theta^* = \arg\min_\theta \mathbb{E}_{(x,y)\sim\mathcal{D}} \left[ \max_{\delta\in\Delta} \mathcal{L}_\theta(x+\delta, y) \right]. \tag{2}$$

This robust objective is in fact common in the context of machine learning security, where $\Delta$ is usually chosen to be a simple convex set, e.g., an $\ell_p$-ball. Canonical instantiations of robust optimization such as adversarial training (Goodfellow et al., 2015; Madry et al., 2018)) have arisen as practical ways of obtaining networks that are invariant to small $\ell_p$-bounded changes in the input while maintaining high accuracy (though a small tradeoff between robustness and accuracy has been noted by prior work (Tsipras et al., 2019; Su et al., 2018) (also cf. Appendix Tables 4 and 5 for a comparison of accuracies of standard and robust classifiers)).

### 3.2 ROBUST TRAINING AS A FEATURE PRIOR

Traditionally, adversarial robustness in the deep learning setting has been explored as a goal predominantly in the context of ML security and reliability (Biggio & Roli, 2018).

In this work, we consider an alternative perspective on adversarial robustness—we cast it as a prior on the features that can be learned by a model. Specifically, models trained with objective (2) must be *invariant* to a set of perturbations $\Delta$. Thus, selecting $\Delta$ to be a set of perturbations that humans are robust to (e.g., small $\ell_p$-norm perturbations) results in models that share more invariances with (and thus are encouraged to use similar features to) human perception. Note that incorporating human-selected priors and invariances in this fashion has a long history in the design of ML models—convolutional layers, for instance, were introduced as a means of introducing an invariance to translations of the input (Fukushima, 1980).

In what follows, we will explore the effect of the prior induced by adversarial robustness on models' learned representations, and demonstrate that representations learned by adversarially robust models

are significantly better-behaved, and enable many previously-infeasible modes of direct interaction with feature embeddings. It is worth noting that despite the value of $\varepsilon$ used for training being quite small, we find that robust optimization *globally* affects the behavior of learned representations. As we demonstrate in this section, the benefits of robust representations extend to out-of-distribution inputs and far beyond $\varepsilon$-balls around the training distribution.

### 3.3 STANDARD AND ROBUST REPRESENTATIONS

Our work is primarily focused on studying the *representations* of trained neural networks. Throughout this work, we define the representation function $R(\cdot)$ as a function induced by a neural network which maps inputs $x \in \mathbb{R}^n$ to vectors $R(x) \in \mathbb{R}^k$ in the representation layer of that network (the penultimate layer). In what follows, we refer to "standard representations" as the representation functions induced by standard (non-robust) networks, trained with the objective (1)—analogously, "robust representations" refer to the representation functions induced by $\ell_2$-adversarially robust networks, i.e. networks trained with the objective (2) with $\Delta$ being the $\ell_2$ ball:

$$\theta^*_{robust} = \arg\min_{\theta} \mathbb{E}_{(x,y)\sim\mathcal{D}} \left[ \max_{\|\delta\|_2 \leq \varepsilon} \mathcal{L}_{\theta}(x + \delta, y) \right].$$

## 4 EXPERIMENTAL SETUP

We train robust and standard ResNet-50 (He et al., 2016) networks on the Restricted ImageNet (Tsipras et al., 2019) and ImageNet (Russakovsky et al., 2015) datasets. Datasets specifics are in in Appendix A.1, training details are in in Appendices A.2 and A.3, and the performance of each model is reported in Appendix A.4. In the main text, we present results for Restricted ImageNet, and link to (nearly identical) results for ImageNet in Appendices (B.1.4,B.3.2).

Unless explicitly noted otherwise, our optimization method of choice for any objective function will be (projected) gradient descent (PGD), a first-order method which is known to be highly effective for minimizing neural network-based loss functions for both standard and adversarially robust neural networks (Athalye et al., 2018a; Madry et al., 2018).

## 5 INVERTING ROBUST REPRESENTATIONS

A common tool for understanding the semantic content of a feature representation is *inversion*—i.e., reconstructing an image using only data from its feature embedding (Mahendran & Vedaldi, 2015; Dosovitskiy & Brox, 2016b; Ulyanov et al., 2017). As discussed in the introduction, however, for standard deep networks, given any input $x$, it is straightforward to find another input that looks entirely different but has nearly the same representation (c.f. Figure 1). This makes representation inversion a difficult problem, since images are not identifiable from their feature representations alone. Indeed, the "naive" approach to representation inversion, wherein one inverts $x$ by solving

$$x_{inv} = x_0 + \arg\min_{\delta} \|R(x_0 + \delta) - R(x)\|_2, \tag{3}$$

tends to find solutions $x_{inv}$ that bear very little resemblance to $x$ (Mahendran & Vedaldi, 2015), and instead look visually similar to the starting point of the optimization $x_0$ (which can be any arbitrary image or even random noise).

**Related work and motivation.** As a result of the phenomenon described above, methods for inverting learned representations typically either impose a "natural image" prior on the inverted image (Mahendran & Vedaldi, 2015; Yosinski et al., 2015; Ulyanov et al., 2017), or train a separate model altogether to perform the inversion (Kingma & Welling, 2015; Dosovitskiy & Brox, 2016b;a). In either case, however, these methods introduce *additional information* into the inversion process beyond that which is contained in feature embedding, making the inversions not fully faithful to the model. In particular, it becomes difficult to disentangle the information that is actually contained in models' feature embeddings from that which is imposed externally by the inversion method.

More broadly, lack of invertibility runs somewhat counter to the idea that learned representations capture a relevant set input features. After all, if the representation function was truly extracting "high-level" features of the input as we conceptualize them, semantically dissimilar images should (by definition) have different representations. We now show that the state of affairs is greatly improved for robust representations.

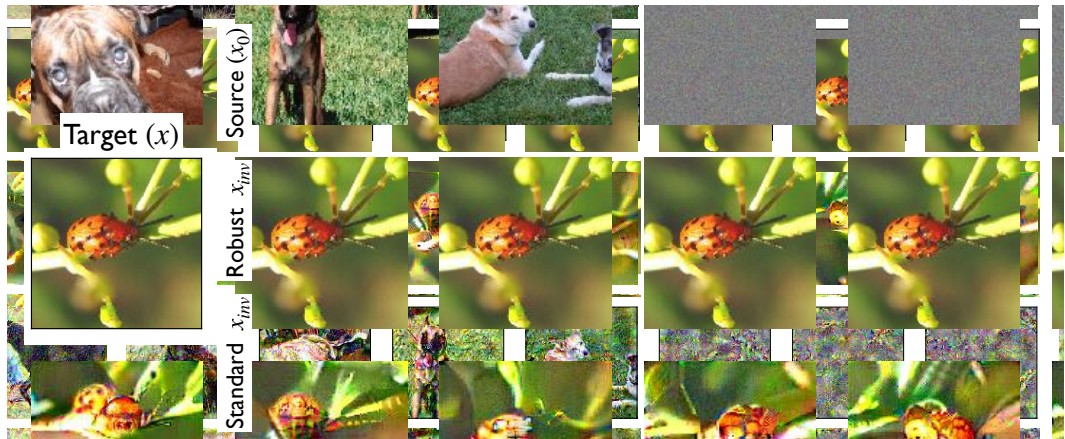

Figure 3: Representation inversions for robust and standard models trained on the Restricted ImageNet dataset. We minimize (3) to find an $x_{inv}$ which matches (in $\ell_2$-distance) the representation of the target image $(x)$ starting from each corresponding source image $x_0$ (top row) for an adversarially trained (second row) and standard (third row) model respectively. When inverting the robust representation, regardless of the optimization starting point $x_0$, the resulting inversions are perceptually similar to the target image; in contrast, the results of inverting the standard model appear more similar to the (arbitrary) source image used as the seed for the optimization. Additional results in Appendix B.1, and similar results for ImageNet are in Appendix B.1.4.

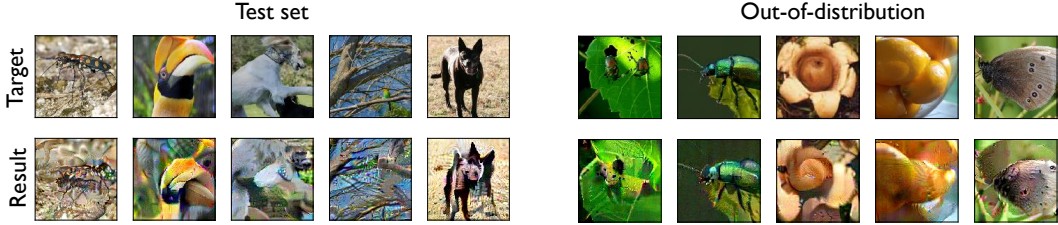

Figure 4: *Target*: random images from the test set (col. 1-5) and from outside of the training distribution (6-10); *Result*: images obtained from optimizing inputs (using Gaussian noise as the source image) to minimize $\ell_2$-distance to the representations of the corresponding image in the top row. More examples appear in Appendix B.1.

**Robust representations are (approximately) invertible out-of-the-box.** It turns out that in sharp contrast to what we observe for standard models, the images resulting from minimizing (3) for robust models are actually *semantically similar* to the original (target) images whose representation is being matched, and this behavior is consistent across multiple samplings of the starting point (source image) $x_1$ (cf. Figure 3).

This inversion property holds even for out-of-distribution inputs, demonstrating that robust representations can represent features beyond those that are relevant for the specific classification task. In particular, we repeat the inversion experiment (simple minimization of distance in representation space) using images from classes not present in the original dataset used during training (Figure 4 right) and structured random patterns (Figure 13 in Appendix B.1): the reconstructed images consistently resemble the targets.

In fact, the contrast between the invertibility of standard and robust representations is even stronger. To illustrate this, we will attempt to match the representation of a target image while staying close to the starting image of the optimization in pixel-wise $\ell_2$-norm (this is equivalent to putting a norm bound on $\delta$ in objective (3)). With standard models, we can consistently get close to the target image in representation space, without moving far from the source image $x_1$. On the other hand, for robust models, we cannot get close to the target representation while staying close to the source image—this is illustrated quantitatively in Figure 5. This indicates that for robust models, semantic similarity may in fact be necessary for representation similarity (and is not, for instance, merely an artifact of the local robustness induced by robust optimization).

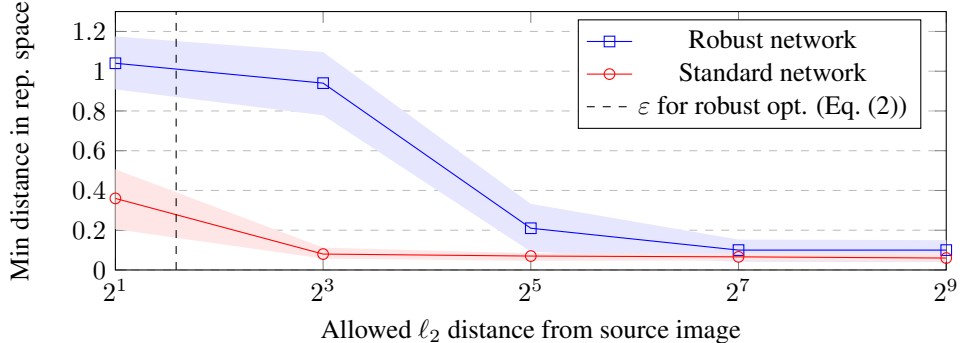

Figure 5: Optimizing objective (3) with an $\ell_2$-norm constraint around the source image. On the $x$-axis is the radius of the constraint set, and on the $y$-axis is the distance in representation space between the minimizer of objective (3) within the constraint set and the target image, normalized by the norm of the representation of the target image: i.e., a point $(x_i, y_i)$ on the graph corresponds to $y_i = \min_{\|\delta\|_2 \leq x_i} \|R(x + \delta) - R(x_{targ})\|_2 / \|R(x_{targ})\|_2$. Notably, we are unable to closely match the representation of the target image for the robust network until the norm constraint grows very large, and in particular much larger than the norm of the perturbation that the model is trained to be robust against ($\varepsilon$ in objective (2)). Shown are 95% confidence intervals over random choice of source and target images.

**Additional related work.** We have already discussed related work in inverting standard representations. In an orthogonal direction, it is possible to construct models that are analytically invertible by construction (Dinh et al., 2014; 2017; Jacobsen et al., 2018; Behrmann et al., 2018). However, the representations learned by these models do not seem to be perceptually meaningful (for instance, interpolating between points in the representation space does not lead to perceptual input space interpolations (Jacobsen et al., 2018), in contrast to robust representations, cf. Appendix A.5.). Another important distinction between the inversions shown here and invertible networks is that the latter are an exactly invertible map from $\mathbb{R}^d \rightarrow \mathbb{R}^d$, while the former shows that we can approximately recover the original input in $\mathbb{R}^d$ from a "compressed" representation in $\mathbb{R}^k$ for $k \ll d$.

## 6 DIRECT FEATURE VISUALIZATION

In the previous section, we showed how adversarial training enables direct (approximate) input inversion without including extra information at inversion-time. We now explore another common technique for studying representations (namely, feature visualization (Olah et al., 2017)) and show that here too, robust representations behave significantly better than their standard counterparts.

In *optimization-based feature visualization* (Olah et al., 2017), we maximize a specific feature (component) in the representation with respect to the input, in order to obtain insight into the role of the feature in classification. Concretely, given some $i \in [k]$ denoting a component of the representation vector, we use gradient descent to find an input $x'$ that maximally activates it, i.e., we solve:

$$x_{vis} = x_0 + \arg \max_{\delta} R(x_0 + \delta)_i \qquad (4)$$

for various starting points $x_0$ which might be random images from $\mathcal{D}$ or even random noise.

**Visualization "fails" for standard networks.** For standard networks, optimizing the objective (4) often yields unsatisfying results. While we *can* easily find images for which the $i^{th}$ component of $R(\cdot)$ is large (and thus the optimization problem is tractable), these images tends to look meaningless to humans, often resembling the starting point of the optimization. Even when these images are non-trivial, they tend to contain abstract, hard-to-discern patterns (c.f. Figure 6 (bottom)). As a result, just as was the case for representation inversion, state-of-the-art feature visualization methods typically regularize objective (4) with terms that encourage more compelling visualizations. These methods include applying random transformations during the optimization process (Mordvintsev et al., 2015; Olah et al., 2017), leveraging deep generative models (Nguyen et al., 2015; 2016; 2017), or post-processing the input or gradients (Oygard, 2015; Tyka, 2016).

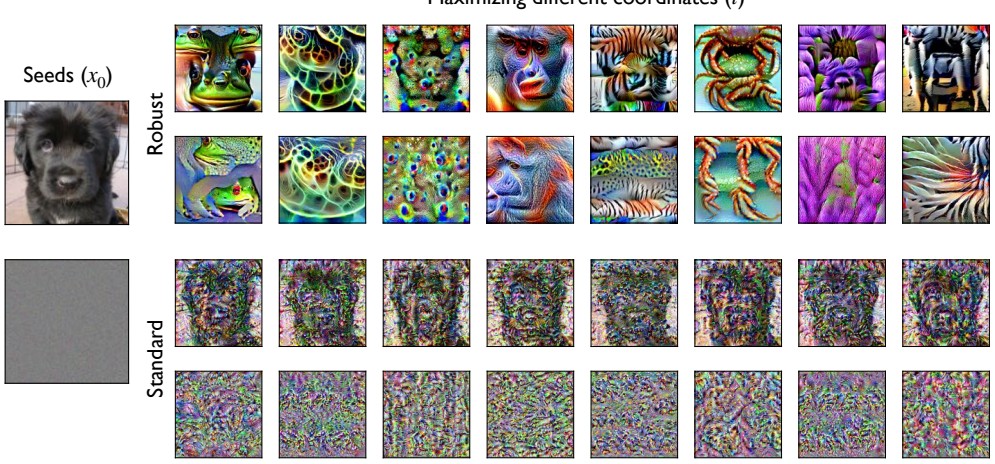

Figure 6: Correspondence between image-level patterns and activations learned by standard and robust models on the Restricted ImageNet dataset. Starting from randomly chosen seed inputs (noise/images), we use PGD to find inputs that (locally) maximally activate a given component of the representation vector (cf. Appendix A.6.1 for details). In the left column we have the seed inputs $x_0$ (selected *randomly*), and in subsequent columns we visualize the result of the optimization (4), i.e., $x'$, for different activations, with each row starting from the same (far left) input $x_0$ for (*top*): a robust (adversarially trained) and (*bottom*): a standard model. Additional visualizations in Appendix B.3, and similar results for ImageNet in B.3.2.

As was the case for inversion, however, such regularization comes with a few well-known (and widely-studied within the feature visualization literature) disadvantages. First, when one introduces prior information about what makes images visually appealing into the optimization process, it becomes difficult to disentangle the effects of the actual model from the effect of the prior information introduced through regularization[2]. Furthermore, while adding regularization does improve the visual quality of the visualizations, the components of the representation still cannot be shown to correspond to any recognizable high-level feature. Indeed, Olah et al. (2017) note that in the representation layer of a standard GoogLeNet, "Neurons do not seem to correspond to particularly meaningful semantic ideas"—the corresponding feature visualizations are reproduced in Figure 7.

**Robust representations allow for direct visualization of human-recognizable features.** For robust representations, however, we find that easily recognizable high-level features emerge from optimizing objective (4) directly, *without any regularization or post-processing*. We present the results of this maximization in Figure 6 (top): coordinates consistently represent the same concepts across different choice of starting input $x_0$ (both in and out of distribution). Furthermore, these concepts are not merely an artifact of our visualization process, as they consistently appear in the test-set inputs that most strongly activate their corresponding coordinates (Figure 9).

---

[2]In fact, model explanations that enforce priors for purposes of visual appeal have been often found to have little to do with the data or the model itself (Adebayo et al., 2018).

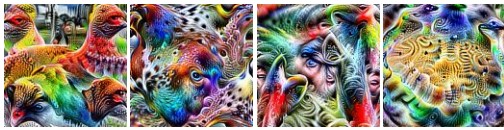

Figure 7: Figure reproduced from (Olah et al., 2017)—a (strongly regularized) visualization of a few components of the representation layer of GoogLeNet. While regularization (as well as Fourier parameterization and colorspace decorrelation) yields visually appealing results, the visualization does not reveal consistent semantic concepts. The situation is significantly worse without regularization (cf. Figure 6 bottom).

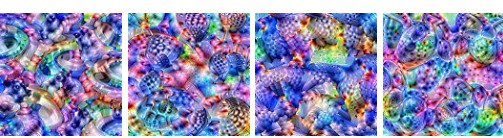

Figure 8: A visualization of the first four components of the representation layer of VGG16 when regularization via random jittering and rotation is applied. Figure produced using the Lucid[a] visualization library.

___________________________

[a]https://github.com/tensorflow/lucid/

In fact, we observe that the *unregularized* feature visualizations from robust representations are significantly more discernable than even *regularized* visualizations of standard ones. Specifically, we provide examples of representation-layer visualizations for VGG16 (which we found qualitatively best among modern architectures) regularized with jittering and random rotations in Figure 8. While these visualizations certainly look better qualitatively than their unregularized counterparts in Figure 6 (bottom), there remains a large gap in quality and discernability between these regularized visualizations and the unregularized robust visualizations in Figure 6 (top).

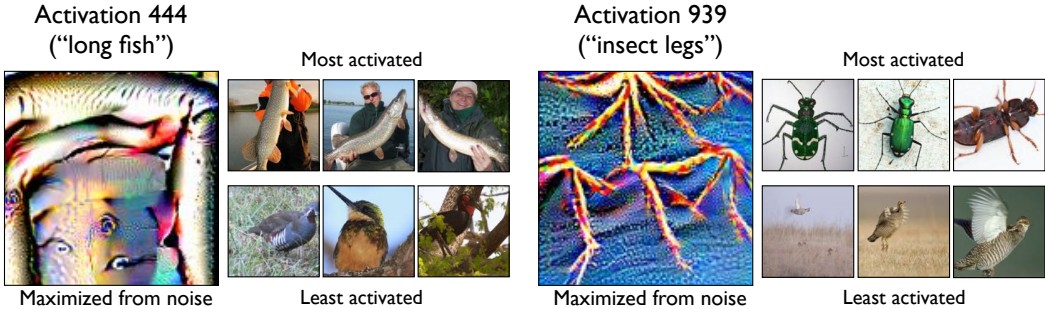

Figure 9: Maximizing inputs $x'$ (found by solving (4) with $x_0$ being a gray image) and most or least activating images (from the test set) for two *random* activations of a robust model trained on the Restricted ImageNet dataset. For each activation, we plot the three images from the validation set that had the highest or lowest activation value sorted by the magnitude of the selected activation.

**Natural consequence: feature manipulation.** The ability to directly visualize high-level, recognizable features reveals another application of robust representations, which we refer to as *feature manipulation*. Consider the visualization objective (4) shown in the previous section. Starting from some original image, optimizing this objective results in the corresponding feature being introduced in a continuous manner. It is hence possible to stop this process relatively early to ensure that the content of the original image is preserved. As a heuristic, we stop the optimization process as soon as the desired feature attains a larger value than all the other coordinates of the representation. We visualize the result of this process for a variety of input images in Figure 10, where *"stripes"* or *"red limbs"* are introduced seamlessly into images without any processing or regularization [3].

**Additional related work.** The latent space of generative adversarial networks (GANs) (Goodfellow et al., 2014) tends to allow for "semantic feature arithmetic" (Radford et al., 2016; Larsen et al., 2016) (similar to that in word2vec embeddings (Mikolov et al., 2013)) where one can manipulate salient input features using latent space manipulations. In a similar vein, one can utilize an image-to-image translation framework to perform such manipulation (e.g. transforming horses to zebras), although this requires a task-specific dataset and model (Zhu et al., 2017). I t is possible to utilize the

___________________________

[3]We repeat this process with many additional random images and random features in Appendix B.4.

Figure 10: Visualization of the results from maximizing a chosen (left) and a *random* (right) representation coordinate starting from *random* images for the Restricted ImageNet dataset. In each figure, the top row has the initial images, and the bottom row has a feature added. Additional examples in Appendix B.4.

deep representations of *standard* models to perform semantic feature manipulations; however such methods tend to either only perform well on datasets where the inputs are center-aligned (Upchurch et al., 2017), or are restricted to a small set of manipulations (Gatys et al., 2016).

## 7 CONCLUSION

We show that the learned representations of robustly trained models align much more closely with our idealized view of neural network embeddings as extractors of high-level features. After highlighting certain shortcomings of standard deep networks and their representations, we demonstrate that robust optimization can actually be viewed as inducing a *prior* over the features that models are able to learn. In this way, one can view the *robust representations* that result from this prior as feature extractors that are more aligned with human perception.

In support of this view, we demonstrate that robust representations overcome the challenges identified for standard representations: they are approximately invertible, and moving towards an image in representation space seems to entail recovering salient features of that image in pixel space. Furthermore, we show that robust representations can be directly visualized with first-order methods without the need for post-processing or regularization, and also yield much more human-understandable features than standard models (even when they are visualized with regularization). These two properties (inversion and direct feature visualization), in addition to serving as illustrations of the benefits of robust representations, also enable direct modes of input manipulation such as feature addition.

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

## A  EXPERIMENTAL SETUP

In this section we describe the elements of our experimental setup. Code for reproducing our results using the `robustness` library (Engstrom et al., 2019) can be found at `https://github.com/snappymanatee/robust-learned-representations`.

### A.1  DATASETS

In the main text, we perform all our experimental analysis on the Restricted ImageNet dataset (Tsipras et al., 2019) which is obtained by grouping together semantically similar classes from ImageNet into 9 super-classes shown in Table 1. Attaining robust models for the complete ImageNet dataset is known to be challenging, both due to the hardness of the learning problem itself, as well as the computational complexity.

For the sake of completeness, we also replicate our experiments feature visualization and representation inversion on the complete ImageNet dataset (Russakovsky et al., 2015) in Appendices B.3.2 and B.1.4—in particular, cf. Figures 21 and 15.

Table 1: Classes used in the Restricted ImageNet model. The class ranges are inclusive.

| Class | Corresponding ImageNet Classes |
|---|---|
| "Dog" | 151 to 268 |
| "Cat" | 281 to 285 |
| "Frog" | 30 to 32 |
| "Turtle" | 33 to 37 |
| "Bird" | 80 to 100 |
| "Primate" | 365 to 382 |
| "Fish" | 389 to 397 |
| "Crab" | 118 to 121 |
| "Insect" | 300 to 319 |

### A.2  MODELS

We use the ResNet-50 architecture (He et al., 2016) for our adversarially trained classifiers on all datasets. Unless otherwise specified, we use standard ResNet-50 classifiers trained using empirical risk minimization as a baseline in our experiments. Additionally, it has been noted in prior work that among standard classifiers, VGG networks (Simonyan & Zisserman, 2015) tend to have better-behaved representations and feature visualizations (Mordvintsev et al., 2018). Thus, we also compare against standard VGG16 networks in the subsequent appendices. All models are trained with data augmentation, momentum $0.9$ and weight decay $5e^{-4}$. Other hyperparameters are provided in Tables 2 and 3.

The exact procedure used to train robust models along with the corresponding hyperparameters are described in Section A.3. For standard (not adversarially trained) classifiers on the complete 1k-class ImageNet dataset, we use pre-trained models provided in the PyTorch repository[4].

Table 2: Standard hyperparameters for the models trained in the main paper.

| Dataset | Model | Arch. | Epochs | LR | Batch Size | LR Schedule |
|---|---|---|---|---|---|---|
| Restricted ImageNet | standard | ResNet-50 | 110 | 0.1 | 256 | Drop by 10 at epochs $\in [30, 60]$ |
| Restricted ImageNet | robust | ResNet-50 | 110 | 0.1 | 256 | Drop by 10 at epochs $\in [30, 60]$ |
| ImageNet | robust | ResNet-50 | 110 | 0.1 | 256 | Drop by 10 at epochs $\in [100]$ |

---

[4]https://pytorch.org/docs/stable/torchvision/models.html

Test performance of all the classifiers can be found in Section A.4. Specific parameters used to study the properties of learned representations are described in Section A.6.

## A.3 ADVERSARIAL TRAINING

To obtain robust classifiers, we employ the adversarial training methodology proposed in (Madry et al., 2018). Specifically, we train against a projected gradient descent (PGD) adversary with a normalized step size, starting from a random initial perturbation of the training data. We consider adversarial perturbations in $\ell_2$-norm. Unless otherwise specified, we use the values of $\epsilon$ provided in Table 3 to train/evaluate our models (the images themselves lie in the range $[0, 1]$).

Table 3: Hyperparameters used for adversarial training.

| Dataset | $\epsilon$ | # steps | Step size |
|---|---|---|---|
| Restricted ImageNet | 3.0 | 7 | 0.5 |
| ImageNet | 3.0 | 7 | 0.5 |

## A.4 MODEL PERFORMANCE

Standard test performance for the models used in the paper are presented in Table 4 for the Restricted ImageNet dataset and in Table 5 for the complete ImageNet dataset.

Additionally, we report adversarial accuracy of both standard and robust models. Here, adversarial accuracies are computed against a PGD adversary with 20 steps and step size of $0.375$. (We also evaluated against a stronger adversary using more steps (100) of PGD, however this had a marginal effect on the adversarial accuracy of the models.)

Table 4: Test accuracy for standard and robust models on the Restricted ImageNet dataset.

| Model | Standard | Adversarial (eps=3.0) |
|---|---|---|
| Standard VGG16 | 98.22% | 2.17% |
| Standard ResNet-50 | 98.01% | 4.74% |
| Robust ResNet-50 | 92.39% | 81.91% |

Table 5: Top-1 accuracy for standard and robust models on the ImageNet dataset.

| Model | Standard | Adversarial (eps=3.0) |
|---|---|---|
| Standard VGG16 | 73.36% | 0.35% |
| Standard ResNet-50 | 76.13% | 0.13% |
| Robust ResNet-50 | 57.90% | 35.16% |

## A.5 IMAGE INTERPOLATIONS

A natural consequence of the "natural invertibility" property of robust representations is the ability to synthesize natural interpolations between any two inputs $x_1, x_2 \in \mathbb{R}^n$. In particular, given two images $x_1$ and $x_2$, we define the $\lambda$-*interpolate* between them as

$$x_\lambda = \min_x \| (\lambda \cdot R(x_1) + (1 - \lambda) \cdot R(x_2)) - R(x) \|_2. \tag{5}$$

where, for a given $\lambda$, we find $x_\lambda$ by solving (5) with projected gradient descent. Intuitively, this corresponds to linearly interpolating between the points in representation space and then finding a point in image space that has a similar representation. To construct a length-$(T + 1)$ interpolation,

we choose $\lambda = \{0, \frac{1}{T}, \frac{2}{T}, \ldots 1\}$. The resulting interpolations, shown in Figure 11, demonstrate that the $\lambda$-interpolates of robust representations correspond to a meaningful feature interpolation between images. (For standard models constructing meaningful interpolations is impossible due to the brittleness identified in Section **??**—see Appendix B.1.3 for details.)

**Top: Image-space interpolation**

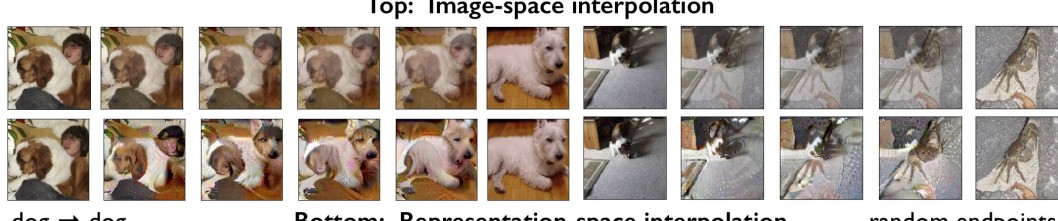

dog → dog      **Bottom: Representation-space interpolation**      random endpoints

Figure 11: Image interpolation using robust representations compared to their image-space counterparts. The former appear perceptually plausible while the latter exhibit ghosting artifacts. For pairs of images from the Restricted ImageNet test set, we solve (5) for $\lambda$ varying between zero and one, i.e., we match linear interpolates in representation space. Additional interpolations appear in Appendix B.2.1 Figure 16. We demonstrate the ineffectiveness of interpolation with standard representations in Appendix B.2.2 Figure 17.

**Relation to other interpolation methods.** We emphasize that linearly interpolating in robust representation space works for *any* two images. This generality is in contrast to interpolations induced by GANs (e.g. (Radford et al., 2016; Brock et al., 2019)), which can only interpolate between images generated by the generator. (Reconstructions of out-of-range images tend to be decipherable but rather different from the originals (Bau et al., 2019).) It is worth noting that even for models with analytically invertible representations, interpolating in representation space does not yield semantic interpolations (Jacobsen et al., 2018).

## A.6 PARAMETERS USED IN STUDIES OF ROBUST/STANDARD REPRESENTATIONS

### A.6.1 FINDING REPRESENTATION-FEATURE CORRESPONDENCE

| Dataset | $\epsilon$ | # steps | Step size |
|---|---|---|---|
| Restricted ImageNet/ImageNet | 1000 | 200 | 1 |

### A.6.2 INVERTING REPRESENTATIONS AND INTERPOLATIONS

| Dataset | $\epsilon$ | # steps | Step size |
|---|---|---|---|
| Restricted ImageNet/ImageNet | 1000 | 10000 | 1 |

# B OMITTED FIGURES

## B.1 INVERTING REPRESENTATIONS

### B.1.1 RECOVERING TEST SET IMAGES USING ROBUST REPRESENTATIONS

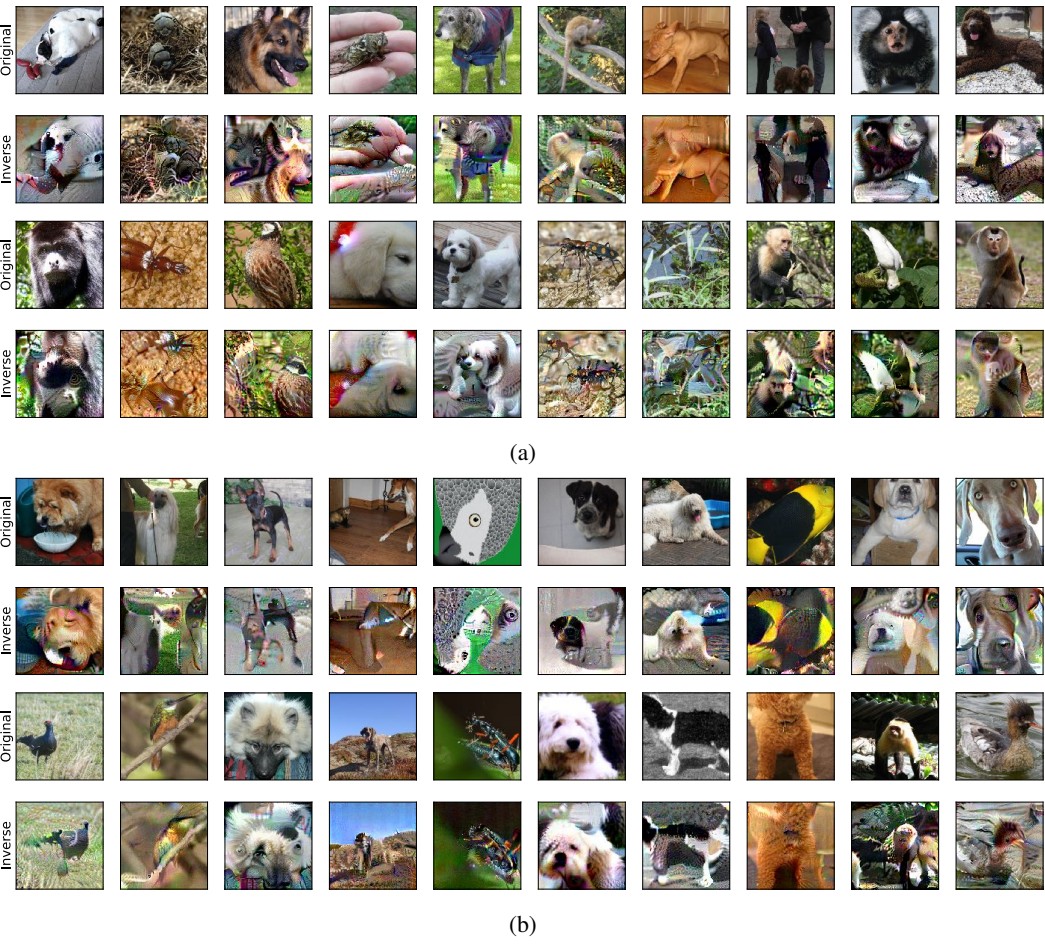

Figure 12: Robust representations yield semantically meaningful inverses: *Original*: randomly chosen test set images from the Restricted ImageNet dataset; *Inverse*: images obtained by inverting the representation of the corresponding image in the top row by solving the optimization problem (**??**) starting from: (a) different test images and (b) Gaussian noise.

### B.1.2 RECOVERING OUT-OF-DISTRIBUTION INPUTS USING ROBUST REPRESENTATIONS

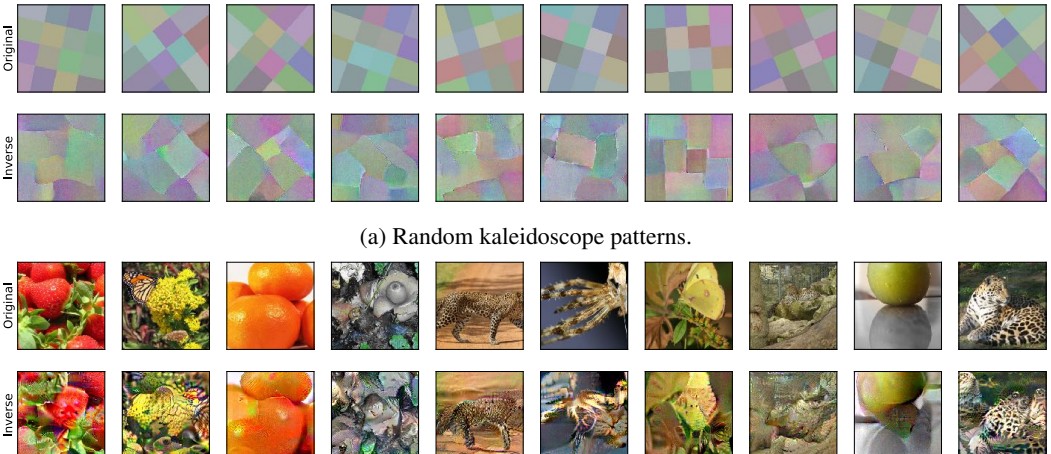

(a) Random kaleidoscope patterns.

(b) Samples from other ImageNet classes outside what the model is trained on.

Figure 13: Robust representations yield semantically meaningful inverses: (*Original*): randomly chosen out-of-distribution inputs; (*Inverse*): images obtained by inverting the representation of the corresponding image in the top row by solving the optimization problem (**??**) starting from Gaussian noise.

### B.1.3 INVERTING STANDARD REPRESENTATIONS

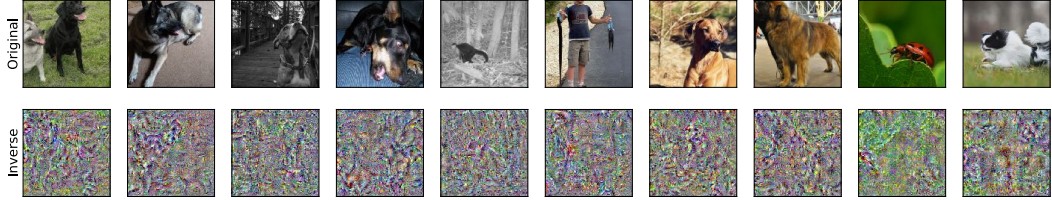

Figure 14: Standard representations *do not* yield semantically meaningful inverses: (*Original*): randomly chosen test set images from the Restricted ImageNet dataset; (*Inverse*): images obtained by inverting the representation of the corresponding image in the top row by solving the optimization problem (**??**) starting from Gaussian noise.

### B.1.4 REPRESENTATION INVERSION ON THE IMAGENET DATASET

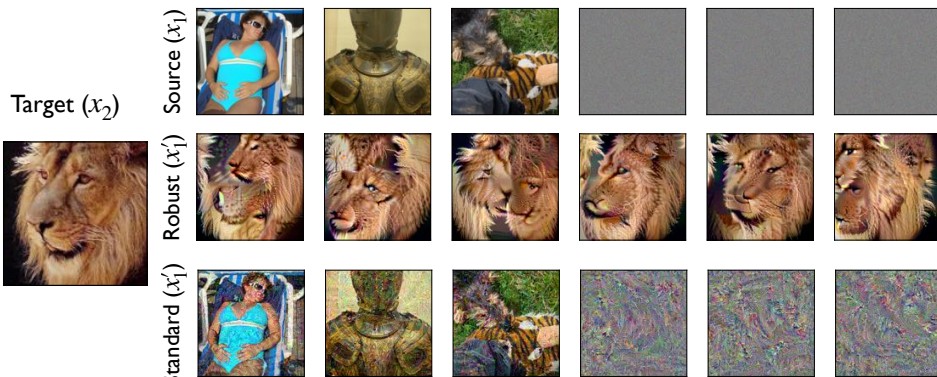

Figure 15: Visualization of inputs that are mapped to similar representations by models trained on the ImageNet dataset. *Target ($x_2$) & Source ($x_1$)*: random examples image from the test set; *Robust* and *Standard ($x_1'$)*: result of minimizing the objective (3) to match (in $\ell_2$-distance) the representation of the target image starting from the corresponding source image for (*top*): a robust (adversarially trained) and (*bottom*): a standard model respectively. For the robust model, we observe that the resulting images are perceptually similar to the target image in terms of high-level features, while for the standard model they often look more similar to the source image which is the seed for the optimization process.

## B.2 IMAGE INTERPOLATIONS

### B.2.1 ADDITIONAL INTERPOLATIONS FOR ROBUST MODELS

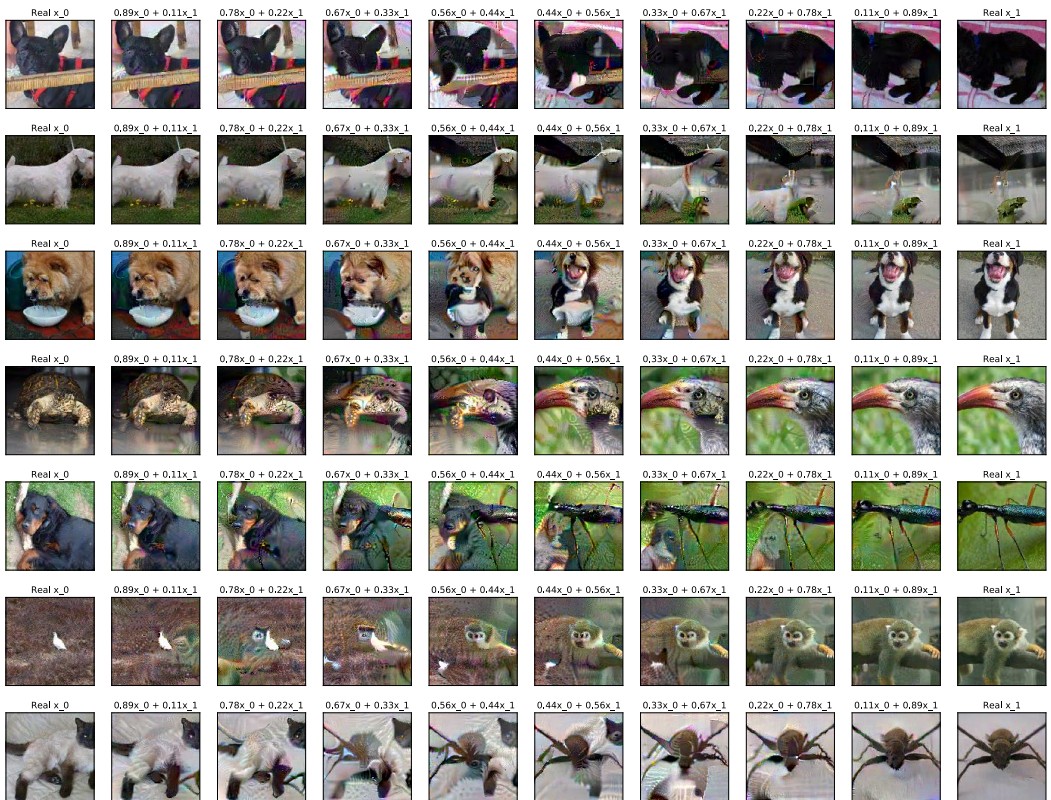

Figure 16: Additional image interpolation using robust representations. To find the interpolation in input space, we construct images that map to linear interpolations of the endpoints in robust representation space. Concretely, for randomly selected pairs from the Restricted ImageNet test set, we use (??) to find images that match to the linear interpolates in representation space (5).

### B.2.2 INTERPOLATIONS FOR STANDARD MODELS

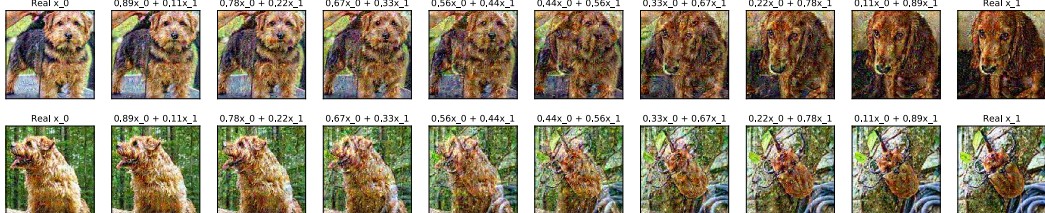

Figure 17: Image interpolation using standard representations. To find the interpolation in input space, we construct images that map to linear interpolations of the endpoints in standard representation space. Concretely, for randomly selected pairs from the Restricted ImageNet test set, we use (**??**) to find images that match to the linear interpolates in representation space (5). Image space interpolations from the standard model appear to be significantly less meaningful than their robust counterparts. They are visibly similar to linear interpolation directly in the input space, which is in fact used to seed the optimization process.

## B.3 Direct feature visualizations for standard and robust models

### B.3.1 Additional feature visualizations for the Restricted ImageNet dataset

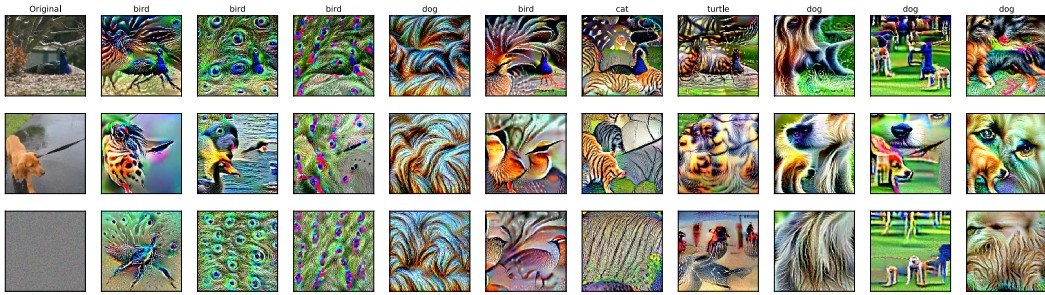

Figure 18: Correspondence between image-level features and representations learned by a robust model on the Restricted ImageNet dataset. Starting from randomly chosen seed inputs (noise/images), we use a constrained optimization process to identify input features that maximally activate a given component of the representation vector (cf. Appendix A.6.1 for details). Specifically, (*left column*): inputs to the optimization process, and (*subsequent columns*): features that activate randomly chosen representation components, along with the predicted class of the feature.

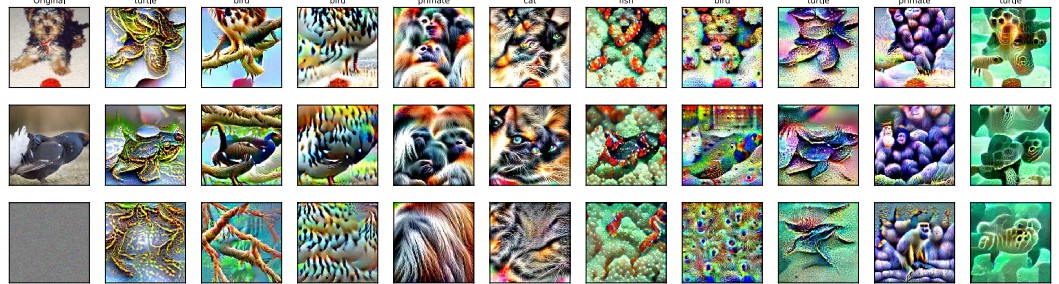

Figure 19: Correspondence between image-level features and representations learned by a robust model on the Restricted ImageNet dataset. Starting from randomly chosen seed inputs (noise/images), we use a constrained optimization process to identify input features that maximally activate a given component of the representation vector (cf. Appendix A.6.1 for details). Specifically, (*left column*): inputs to the optimization process, and (*subsequent columns*): features that activate select representation components, along with the predicted class of the feature.

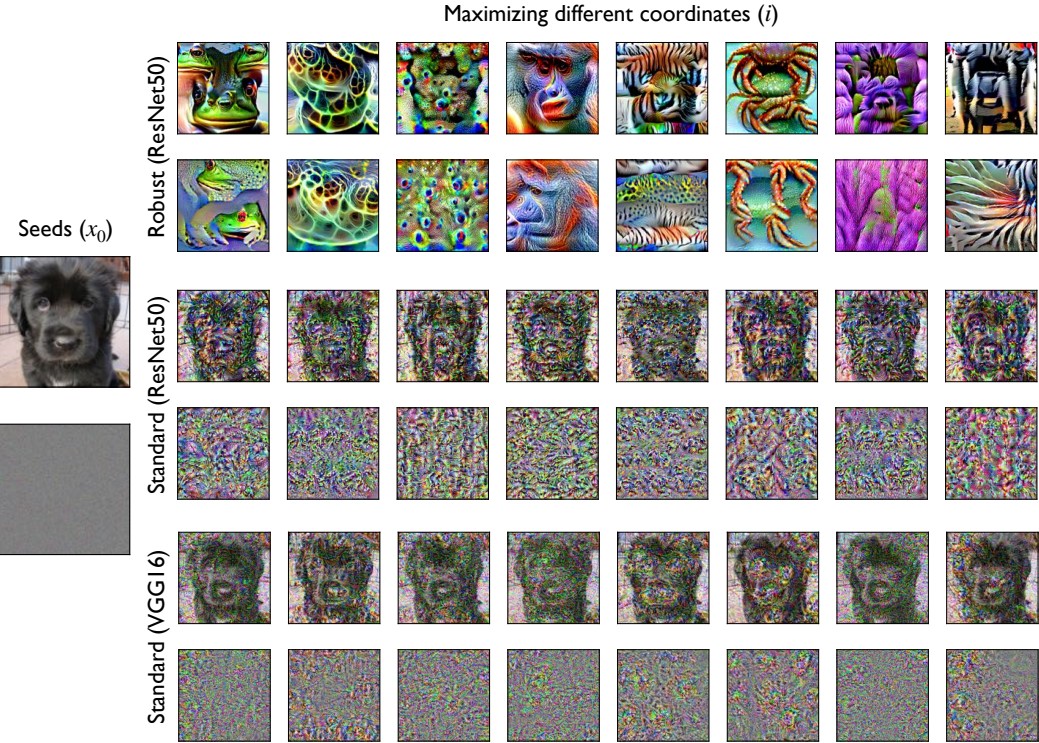

Figure 20: Correspondence between image-level patterns and activations learned by standard and robust models on the Restricted ImageNet dataset. Starting from randomly chosen seed inputs (noise/images), we use PGD to find inputs that (locally) maximally activate a given component of the representation vector (cf. Appendix A.6.1 for details). In the left column we have the original inputs (selected *randomly*), and in subsequent columns we visualize the result of the optimization (4) for different activations, with each row starting from the same (far left) input for (*top*): a robust (adversarially trained) ResNet-50 model, (*middle*): a standard ResNet-50 model and (*bottom*): a standard VGG16 model.

### B.3.2 Feature visualizations for the ImageNet dataset

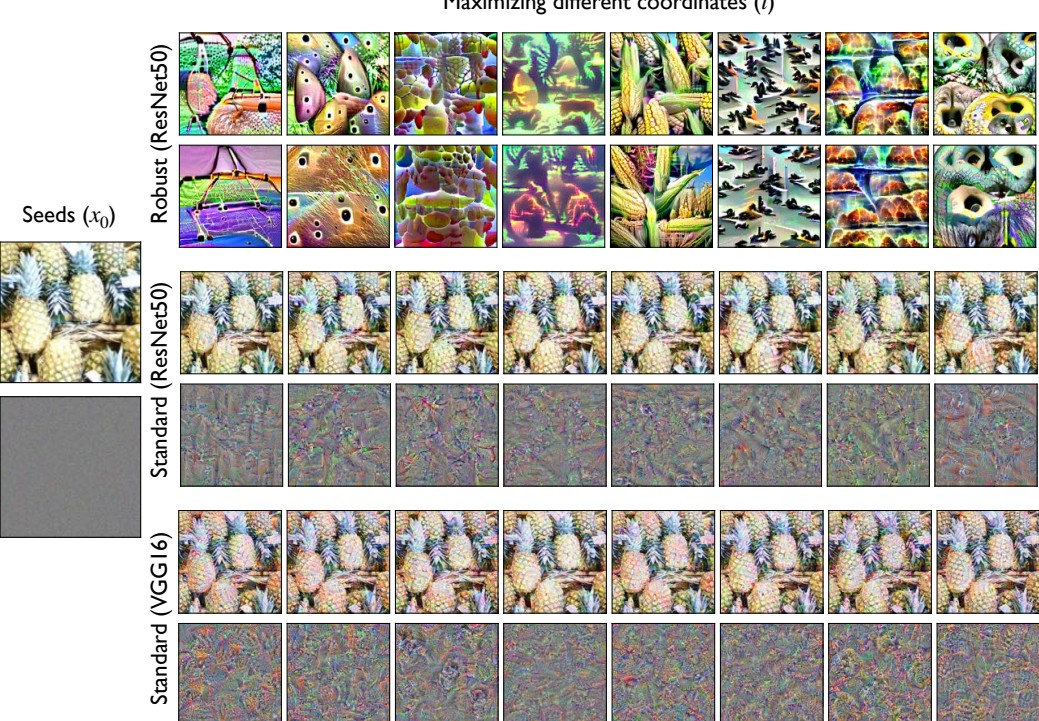

Figure 21: Correspondence between image-level patterns and activations learned by standard and robust models on the complete ImageNet dataset. Starting from randomly chosen seed inputs (noise/images), we use PGD to find inputs that (locally) maximally activate a given component of the representation vector (cf. Appendix A.6.1 for details). In the left column we have the original inputs (selected *randomly*), and in subsequent columns we visualize the result of the optimization (4) for different activations, with each row starting from the same (far left) input for (*top*): a robust (adversarially trained) ResNet-50 model, (*middle*): a standard ResNet-50 model and (*bottom*): a standard VGG16 model.

## B.4 ADDITIONAL EXAMPLES OF FEATURE MANIPULATION

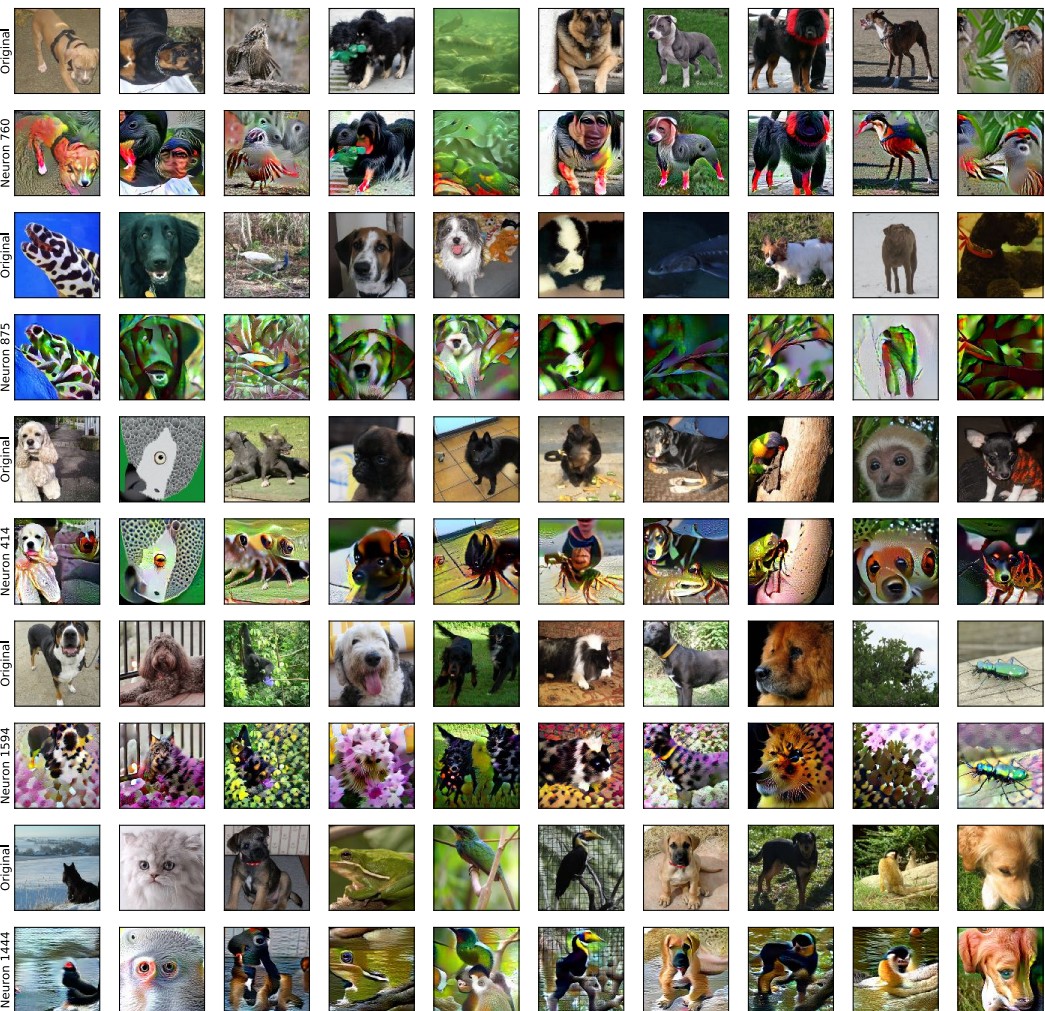

Figure 22: Visualization of the results adding various neurons, labelled on the left, to randomly chosen test images. The rows alternate between the original test images, and those same images with an additional feature arising from maximizing the corresponding neuron.

