# OpenReview forum: "Adversarial Robustness as a Prior for Learned Representations"
_ICLR.cc/2022/Conference — ICLR 2022 Submitted_

### Official Review · Reviewer_h5Za · 2021-10-30

**Correctness:** 4
**Technical Novelty And Significance:** 2
**Empirical Novelty And Significance:** 2
**Recommendation:** 3
**Confidence:** 4

**Main Review:**

## Strength

- This paper conducts instrumental feature visualization experiments, including representation inversion and feature visualization. These experiments demonstrate that adversarial trained neural networks learn a high-level encoding of inputs.
- This paper's experiments are comprehensive. *Figure 5* is especially interesting since it demonstrates how the feature space of a robust network differs from that of a standard network.

## Weakness and questions

- Tsipras et al. (2019) also finds that robust model's feature are more discernable than those of standard model. Moreoever, They also conducts similar experiments to visualize robust model's features. It would be good if the authors of this paper could explicitly discuss what the different between their work and Tsipras et al. (2019).
- In section 3.2, the authors write *selecting $\Delta$ to be a set of perturbations that humans are robust to*. Can the authors explain why $\ell_2$-norm is chosen? How about $\ell_{\infty}$-norm?
- In the *Robust representations are (approximately) invertible our-of-the-box* subsection, the authors find that *this iversion property holds even for out-of-distribution inputs*. It is unclear why robust networks learn discernable features for *OOD* data even if the network has never observed those *OOD* data during training?
- In *Figure 5*, 95% confidence intervals are shown. It would be good if the authors could explain what the 95% confidence interval means and how it is computed.

**Summary Of The Paper:**

This paper empirically demonstrates that robust optimization encourages deep neural networks to learn a high-level encoding of inputs. Specifically, this paper first utilize $\ell_2$- norm adversarial training to train robust neural networks. Then, this paper leverage two visualization techniques, i.e., *representation inversion* and *feature visualization*, to demonstrate that features identified by the robust neural network are more discernable.

To summary, the main contributions of this paper are as follows:

- A comprehensive literature review about findings related to features of standard and robust models;
- Train a $\ell_2$-norm robust neural network and visualize its features;
- Leverage *representation inversion* and *feature visualization* to demonstrate the features in robust neural networks are more discernable.

**Summary Of The Review:**

This paper conducts interesting experiments to demonstrate that robust networks' features are more discernable via representation inversion and feature visualization. However, it is unclear to me how this work is different from previous paper (Tsipras et al. (2019)).

---

### Official Review · Reviewer_DfB9 · 2021-11-02

**Correctness:** 4
**Technical Novelty And Significance:** 2
**Empirical Novelty And Significance:** 2
**Recommendation:** 5
**Confidence:** 4

**Main Review:**

I think this paper is very well written and the problem is well motivated. I think strongest point of the paper is how it is able to invert representations (very well) without regularization and starting from diverse starting points.

Here are some of my thoughts/criticisms:
1. I agree with the authors that feature activations seem to sharper via their method (like in Fig 6), but just because it appears sharper to us humans might not necessarily mean they are more interpretable. To their credit, the authors partially answer this in Fig 9 where certain activations seem to correspond to similar looking images. I think that to make their point clear the authors should have mentioned features that have the highest value in a given image (say top 5) and then show the maximally activated images for those features. This way we know that these features truly are important (say top-5) for their entire class  vs important to just a few images in that class. At the moment, Fig 9 seems to me like a post hoc way of analyzing features.
2. In standard models, usually, the motivation to visualize representations seems to be to understand discriminative features in the input. In robust models (which have lower accuracy), the models seem to be retaining a lot of information not relevant to classification. My question is -  is visualizing a representation useful if that representation /classifier is not very good? I’m not sure about the answer myself but I’m curious what the authors think.
3. Santurkar et al. have studied robust models in a very similar setting to the results in Fig 10 (image to image translation, feature painting)
4. This is just a curiosity - I’ve noticed work in this direction (Tsipras et al., Santurkar et al.) choose l2 vs l-inf, is there a reason for this?

**Summary Of The Paper:**

This paper posits that adversarial training imposes a prior on intermediate representations of a classifier. The authors then go on to show that this prior makes tasks like representation inversion and visualization easier.

**Summary Of The Review:**

I really like the simple and elegant nature of the solution proposed in this paper but, overall, I’m not convinced about the novelty wrt prior work, so I’m going recommend a borderline reject (however, I’m willing to be persuaded)

---

### Official Review · Reviewer_Ukbn · 2021-11-02

**Correctness:** 3
**Technical Novelty And Significance:** 3
**Empirical Novelty And Significance:** 3
**Recommendation:** 5
**Confidence:** 2

**Main Review:**

## Strengths
1. The paper is very well-written and well-organized. The observations are interesting and inspiring.

## Weaknesses
1. My main concern is the novelty of the paper since the current submission does not introduce a new approach or algorithm or theoretical results. The paper also lacks comparison/discussion of recent works.

**Summary Of The Paper:**

This paper presents an interesting observation that the learned representations of robustly trained models align much more closely with our idealized view of neural network embeddings as extractors of high-level features that are more aligned with human perception. The paper provides experiments to support this view: (1) robust representations are (approximately) invertible out-of-the-box; (2) robust representations allow for direct visualization of human-recognizable features.

**Summary Of The Review:**

Reviewing this paper gives me the impression that the paper is a little outdated (for example the most recent paper cited was in 2019). I found a very similar submission for ICLR 2020 and I agree with the reviewers/CPC's discussions. Although the paper is very well-presented, the previous concerns have not been fully addressed in the current manuscript. Thus I lean toward rejecting the paper.

---

### Official Review · Reviewer_o96F · 2021-11-02

**Correctness:** 3
**Technical Novelty And Significance:** 2
**Empirical Novelty And Significance:** 2
**Recommendation:** 3
**Confidence:** 5

**Main Review:**

The properties investigated in the paper are interesting but not surprising, especially in the context of existing literature[1,2,3,4].

The paper only provides experimental demonstration of this phenomenon without going into a more detailed explanation of the phenomenon. In my opinion this is not enough when the observations, in question, are not very novel and have already been explored in various forms in past published literature.

For example [1] argues that local lipschitzness provides adversarial robustness (this is not the only paper to argue this) and local lipschitzness is a necessary (and bi-lipschitzness is a suffiicient) criterion for **invertibility**. [2] shows that adversarially robust models captures salient features of the data and combining that with local linearisation (from [1]) allows us  to **visualise  features**. Thus, in the context of [2], [1], and also [3] i find the feature visualisation experiments rather deductive.

Added to this, [4] provides a whole host of experimental evidence that adversarially robust models can be used to generate salient features of the input data either from scratch, inpaint into an occluded image, or  from hand drawn sketches. Thus it is not a surprise that input space features can be visualised by altering the representation space.

Thus given that the observations themselves are quite deductive from existing work, I would have liked to see a deeper dive into why this happens, if this is a property of adversarial training algorithms or perhaps robust models (using other techniques) also exhibit this. It is surprising that there is much mention of adversarial training but no experimental work with (or even mention of) TRADES [5], which is an algorithm of similar flavor but has shown (perhaps) better empirical behavior.

In the discussion of learned representation, I also did not find mention of recent works [6,7]  that talks about the importance of proper prior for representation learning  in the context of adversarial robustness.

Further, all the main claims of the paper have been exposed through visual examples. While this fits quite well as the points to be made are very visual points, it raises questions about whether the study has been done systematically. There is a huge body of work in the invertibility of NN literature. Perhaps, the authors could see if that could be used to derive more quantitative claims about adversarially robust models.



[1] Yang, Yao-Yuan, et al. "Adversarial robustness through local lipschitzness." (2020).
[2] Salman, Hadi, et al. "Do adversarially robust imagenet models transfer better?." (2020).
[3] Ilyas, Andrew, et al. "Adversarial examples are not bugs, they are features." (2019).
[4] Santurkar, Shibani, et al. "Image synthesis with a single (robust) classifier." (2019).
[5] Zhang, Hongyang, et al. "Theoretically principled trade-off between robustness and accuracy."  (2019).
[6] Montasser, Omar, Steve Hanneke, and Nathan Srebro. "Vc classes are adversarially robustly learnable, but only improperly." (2019).
[7] Sanyal, Amartya, et. al. "How Benign is Benign Overfitting?" (2020).
================== Update ====================

The authors have not provided any response to our reviews. So, I will stick to my initial rating and I am confident of it.

**Summary Of The Paper:**

The paper looks at  favorable properties of feature representations of an adversarially robust model. In particular, the authors look at a model trained with PGD training with an $\ell_p$ adversary. In terms of favourable properties, the authors look at representation inversion and feature manipulation and with experimental evidences claim that adversarially robust models are naturally better at it,

**Summary Of The Review:**

Overall, due to the points mentioned above I found the work to be very deductive and not systematic. Thus I vote for rejection.

---

### Decision · Program_Chairs · 2022-01-20

**Decision:**

Reject

**Comment:**

The paper looks at the favorable properties of feature representations of an adversarially robust model, which are interesting but not surprising, especially in the context of much existing literature has talked about this. All reviewers gave negative scores. The main issues are: 1) The paper only provides experimental demonstration of this phenomenon without going into a more detailed explanation of the phenomenon. This is not enough when the observations, in question, are not very novel and have already been explored in various forms in past published literature. 2) limited novelty since the current submission does not introduce a new approach or algorithm or theoretical results. The paper also lacks comparison/discussion of recent works. Thus, I cannot recommend accepting the paper to ICLR.